# Genomic Characterization of Multiple Antibiotic-Resistant *Enterococcus* in Farm Animals in Ningxia Province, China

**DOI:** 10.3390/antibiotics14111137

**Published:** 2025-11-10

**Authors:** Haoyu Zhao, Wen Zhang, Tianran Tang, Likun Zhang, Shengling Cui, Shengli Chen, Huafang Hao, Yating Deng, Weimin Zhang, Qi Yang, Zengqi Yang, Qian Shao, Juan Wang

**Affiliations:** 1Department of Preventive Veterinary Medicine, College of Veterinary Medicine, Northwest Agricultural and Forestry University, Xianyang 712100, China; hyzhao98@nwafu.edu.cn (H.Z.); ttr@nwafu.edu.cn (T.T.); zlk2001@nwafu.edu.cn (L.Z.); ylzhangwm@nwsuaf.edu.cn (W.Z.); 2Key Laboratory for Prevention and Control of Major Ruminant Diseases, Ministry of Agriculture and Rural Affairs, Xianyang 712100, China; 3Ningxia Supervision Institute for Veterinary Drugs and Animal Feedstuffs, Yinchuan 750004, China; nxyczwgalaxy@sina.cn (W.Z.); cuishenlin@163.com (S.C.); nxyt_411@163.com (Y.D.); yangqmz@163.com (Q.Y.); 4Lanzhou Veterinary Research Institute, Chinese Academy of Agricultura Sciences, Lanzhou 730050, China; chenshengli@caas.cn (S.C.); haohuafang@caas.cn (H.H.)

**Keywords:** *Enterococcus*, farmed animals, antibiotics resistance genes, virulence genes, plasmids

## Abstract

Background/Objectives: In an era of increasing bacterial resistance, *Enterococcus*, as a reservoir of antibiotic resistance genes, poses a serious threat to public health. Methods: This study conducted antibiotic susceptibility tests, whole-genome sequencing, and bioinformatics analysis on 89 *Enterococcus* isolates from chickens, pigs, cattle, and sheep in Ningxia Autonomous Region. Results: The resistance rates of *Enterococcus* to clindamycin, cefoxitin, sulfamethoxazole, and tamoxifen were all above 95%, and 96.6% (86/89) of the isolates were multi-antibiotic resistant. There were significant differences in resistance phenotypes among different species, with *Enterococcus* from pigs showing significantly higher resistance than those from other animals. *optrA* was commonly found in *Enterococcus* from pigs, accounting for 61.5% (8/13). ST480, ST16, ST116, and ST300 were the main MLST types, and ST16 was one of the important pathogenic *Enterococcus* types. Conclusions: The study revealed the occurrence of inter-species transmission events of *Enterococcus*. In conclusion, this study comprehensively described the resistance spectrum, sequence characteristics, and transmission features of resistance genes in *Enterococcus* isolated from farm animals, and emphasized the possibility of the spread of resistance genes carried by *Enterococcus* from farm animals to humans.

## 1. Introduction

*Enterococcus* are the natural intestinal flora of humans and animals [1]. Due to their prevalence in human and animal feces and their persistence in the environment, *Enterococcus* are important key indicator bacteria for antibiotics resistance surveillance systems in humans and veterinarians [2,3]. In addition, *Enterococcus* harboring virulence genes are considered opportunistic pathogens, which have become one of the major causes of nosocomial and community-acquired human infections, including sepsis, endocarditis, and urinary tract infections [4,5,6,7]. During the COVID-19 pandemic, the reports of diseases caused by antibiotics-resistant *Enterococcus* have also increased, further increasing the threat of COVID-19 to human life and health [8,9,10].

In addition, antimicrobial resistance genes in *Enterococcus* can be transferred both intra- and intra-species through mobile genetic elements such as plasmids, transposons (e.g., IS1216, Tn1546), and integrative conjugative elements. These mechanisms facilitate horizontal gene transfer and accelerate the dissemination of resistance determinants. Pheromone-responsive plasmids (e.g., rep9 family) and broad-host-range elements facilitate efficient horizontal gene transfer among *Enterococcus* populations and even between genera under selective pressure in livestock environments [11]. There is a risk of transmission to humans through the food chain. At present, due to the extensive use of antibiotics by human doctors and veterinarians, the resistance of *Enterococcus* to vancomycin, macrolides and other antibiotics has increased rapidly [12,13,14]. In 2024, the World Health Organization listed vancomycin-resistant *Enterococcus faecium* as one of the key bacterial pathogens. In-depth analysis of *Enterococcus* resistance genes revealed the frequent presence of various alleles (e.g., *poxtA*, *optrA*, *cfr*) as well as IS (e.g., IS1216), *optrA*, *poxtA* and *cfr* genes as part of a plasmid or as a rotor complex; It is well known that the probability of transferring these genes to other bacteria is very high [15,16]. More research is therefore needed to determine the risk of emergence and transmission of antibiotic-resistant *Enterococcus* from animals to humans.

As a big breeding province in northwest China, Ningxia has a very developed breeding industry. The main breeding animals are pigs, chickens, cattle and sheep. The use of antibiotics is inevitable in the farming process [17]. However, there is no detailed report on the antibiotic resistance and genomic characteristics of *Enterococcus* isolated from farmed animals. In this study, the whole genome sequencing was used to comprehensively analyze the antimicrobial resistance spectrum of *Enterococcus* in Ningxia, including the phylogenetic group multilocus sequence type (MLST), antibiotic resistance genes and plasmid types. Our study provides the first comprehensive overview of the antibiotic resistance profile and genomic sequence characteristics of *Enterococcus* in Ningxia, and highlights the possibility of transmission from farmed animals to humans as spreaders of antibiotic resistance genes.

## 2. Results

### 2.1. Strain Source

A total of 540 strains of *Enterococcus faecalis*, 214 strains of *Enterococcus faecium*, and other *Enterococcus* 754 strains were isolated and identified from 25 large-scale farms in various municipalities of Ningxia, China, from 2019 to 2023 (Appendix A). Randomly selected 89 *Enterococcus* strains were collected from all over Ningxia province, including species including: pig 13, cattle 28, sheep 8 and chicken 40 (Figure 1C). The 89 strains of *Enterococcus* were *E. faecalis* 58, *E. faecium* 19 and *Enterococcus* 11 (Figure 1B). The sampling area covered the whole Ningxia province municipalities the sampling map is shown in (Figure 1A).

### 2.2. Results of Antibiotic Sensitivity Test

The results of the antibiotic sensitivity test showed that the resistance rate of *Enterococcus* isolates to clindamycin, cefoxitin, sulfisoxazole and tamoxifen was over 95%, followed by ceftiofur 87.6%, benzoxiline 87.6%, ofloxacin 84.3%, erythromycin 82.0%, etc., and the rate of vancomycin was more sensitive to vancomycin with a resistance rate of 3.4% (Figure 2A, Table 1).

Resistance analysis for different species showed that the overall resistance level was the highest for *Enterococcus* of porcine origin, followed by chicken, sheep and cattle origin in descending order. We found that the resistance to linezolid was significantly higher in swine-derived *Enterococcus* than in other animal sources, ranging from 10% to 14.2% in other animal sources, whereas the resistance of swine-derived *Enterococcus* to linezolid reached 61.5% (Figure 2C). The resistance phenotypes of *Enterococcus* of porcine origin and *Enterococcus* of chicken origin were also closer to each other. The number of multi-resistant strains was 86, accounting for 96.6% of the total, of which 7 strains were resistant to all 8 classes of antibiotics (Figure 2B, Appendix A).

### 2.3. Results of Characteristic Genes

The virulence genes showed that the bacterial hair manipulator genes *ebpA*, *ebpB* and *ebpC*; the *fsrA*, *fsrB* and *fsrC* genes of the virulence regulator fsr system are the most important virulence genes; the polymorphisms and colonization/invasion virulence genes *srtC* and biofilm-formation-associated virulence *cpsA* were found to be much more prevalent than those of other virulence genes in *Enterococcus* (Figure 3A). Five strains carried the plasmid-transferred virulence gene asa1, which mediates the production of aggregates and plays a key role in the adhesion of *Enterococcus* to renal and endocardial cells, as well as in the invasion of *Enterococcus* [18].

Of all the ARGs analyzed, 27 ARG types were detected in this study, and among these 27 ARG types, we collated the top16 ARGs (Appendix A). these included aminoglycoside resistance genes *aph(3′)-III*, *aac(6′)-II*, *ant(6)-Ia*, macrolide resistance genes *erm(A)*, *erm(B)*, *msr(C)*, tetracycline resistance genes *tet(L)*, *tet(M)*, chloramphenicol resistance genes cat, *fexA*, lincomycin resistance genes *lsa(A)*, *lsa(E)*, linezolid resistance genes *optrA*, streptomycin resistance genes str, methicillin resistance genes *dfrG*, and lincomycin resistance genes *lun(B)* (Figure 3B). Surprisingly, 17 isolates carrying the oprtA gene were found in Ningxia. Linezolid is banned for animal use in China, but the rates of linezolid resistance and *optrA* carriage in *Enterococcus* from pigs in this study were much higher than normal values. Significant associations were observed between *optrA* and linezolid resistance (*p* < 0.05), and between erm(B) and erythromycin resistance (*p* < 0.01).

The analysis according to the source of different species showed that the carrying rate of resistance genes of *Enterococcus* pigs was significantly higher than that of other species (Figure 3C), and there was no significant difference in the number of virulence genes carried among different species (Figure 3D). The results of antibiotic resistance genes were consistent with the results of combined antibiotic susceptibility, suggesting that porcine *Enterococcus* is the main reservoir of antibiotic resistance genes, and we should pay more attention to the antibiotic resistance of porcine *Enterococcus*. The results showed that the number of antibiotic resistance genes (Figure 3E) and virulence genes (Figure 3F) carried by *E. faecalis* was significantly higher than that of other *Enterococcus* species. *E. faecalis* showed the characteristics of high virulence and high antibiotic resistance in this study, which should be paid more attention in the cultivation process.

### 2.4. Results of MLST Sequence Typing

Among the five ST types (i.e., ST506, ST476, ST593, ST179, and ST535), all isolates possessed the *optrA*-encoding gene (Figure 4). A total of 29 strains (35.6%) had combinations of known ST alleles. We further evaluated the clonality and branching of all 89 *Enterococcus* isolates. Our results suggest that resistance genes such as *optrA* are spread horizontally through *Enterococcus* in large numbers.

### 2.5. Phylogenetic Tree of SNP and Characteristics of Antimicrobial Resistance Genes and Plasmids Carried by Enterococcus

SNP phylogenetic analysis of 89 *Enterococcus* strains showed that the whole phylogenetic tree was divided into three branches, and most isolates from the same species clustered together. Species was the main reason for the clustering of *Enterococcus* phylogenetic tree. For example, the difference between NXC77 and NXC53 SNPS was less than 10, which suggested the occurrence of cross-species transmission events in *Enterococcus* (Figure 5). The 89 *Enterococcus* isolates were divided into three branches. Clade C1 was dominated by ST16 and ST116 and mainly carried plasmids rep10, rep9a and rep9b. Clade C2 was dominated by ST480 and mainly carried plasmid rep10. Both clades carried a large number of antibiotic resistance genes. Clade C3 was dominated by *E. faecium* and mainly carried plasmids repUS15 and rep2. The number of antibiotic resistance genes in clade C3 was significantly less than that in clade C1 and C2.

### 2.6. Co-Existence of Plasmids with Antibiotic-Resistant Genes

In 89 *Enterococcus* strains we observed that plasmid rep10 could coexist with all resistance genes, while rep9a, rep9b and rep9c, could coexist with a large number of resistance genes but not with *msr(c)*, *aac(6′)*-li and *aac(6′)*-lid. At the same time, we observed that strains carrying the *optrA* resistance gene contained the plasmids rep9 (Figure 6). This also suggests to us the importance of rep9 for *optrA* transmission.

## 3. Discussion

The origin of antibiotic resistance is still unclear, and studies have reported the emergence of methicillin-resistant Staphylococcus aureus well before methicillin began to be used in the clinic. This suggests that the emergence of bacterial resistance is not necessarily related to the high use of antibiotics. The initial generation of resistance genes does not serve the function of making bacteria resistant, but rather functions such as metabolism, signaling and detoxification [19,20]. The resistance mechanism is an important driver of microbial evolution that has continued from ancient times to the present [1]. However, when humans apply new antibiotics as clinical drugs, this move puts unprecedented selection pressure on microbes. Understanding and controlling the spread and development of bacterial resistance is therefore one of the most urgent tasks for human health management in the 21st century.

*Enterococcus* are important key indicator bacteria for human and veterinary antibiotic resistance surveillance systems [5]. This study revealed the antibiotic resistance profile of *Enterococcus* in Ningxia region, and the results showed that *Enterococcus* exhibited a high level of antimicrobial resistance in Ningxia region, in which *Enterococcus* resistance to clindamycin, cefoxitin, sulfisoxazole, and tamifluorfen was more than 95%. It is noteworthy that most of the strains in this study (86/89, 96.6%) were resistant to antibiotics of class 3 and above, which implies that it is even more difficult to treat the diseases caused by them. There were some differences in strain resistance between species, with the resistance rates of pig- and chicken-origin *Enterococcus* being significantly higher than those of cattle and sheep, which may be related to farming patterns and medication habits. We found that the resistance of *Enterococcus* of porcine origin to linezolid was significantly higher than that of other animal sources, this is in line with the results of the antibiotic-resistant gene. Linezolid is banned for use in animals in China, but 61.5% of *Enterococcus* isolated from pigs carried the linezolid resistance gene *optrA*. This suggests that *optrA* is more prone to spread among *Enterococcus* from pigs. *Enterococcus* isolated from pigs is an important reservoir of antibiotic resistance genes. As an important economic animal, pigs may become an important reservoir of antibiotic resistance genes in the breeding process, and then spread to humans and other organisms through the food chain or other ways.

In this study, *optrA* was unevenly distributed among *Enterococcus* species and plasmid families, with the entire *optrA* present in *E. faecalis* and identified on rep9 variants of the RepA_N plasmid [21], which are narrow host range plasmids thought to be specific for *E. faecalis*. rep9 family plasmids contain several characteristic pheromone responsive plasmids, such as pAD1. Coupling systems based on sex pheromones have been described as very efficient transfer vectors that can facilitate ARGs transfer. All these observations may explain the high detection rate of rep9 replicon in *optrA*-positive *E. faecalis* in the present study, so further investigation of cross-host transmission of this plasmid type is warranted. Previous studies have shown that pheromone-responsive conjugative plasmids, particularly those in the rep9 family, play a central role in mediating the horizontal transfer of *optrA* among *Enterococcus* spp. Such plasmids contain mobilization genes and sex pheromone systems that enable efficient intra- and interspecies gene transfer, especially under selective pressure from phenicols or macrolides. These mechanisms likely explain the high frequency of *optrA*-rep9 associations observed in our isolates and their potential contribution to ARG dissemination in livestock environments [22,23].

The high resistance rate in the *optrA*-positive group in this study was observed in multiple types of farm samples in China. Our results indicated not only a high resistance rate but also a large number of ARGs in the *optrA*-positive *Enterococcus*, indicating that *optrA*-positive *Enterococcus* pose a higher risk to public health [24,25]. Previous studies have shown that increased daily pork consumption and hospitalization are associated with increased risk of *optrA* carriage in these healthy populations [26]. Despite the heterogeneity in population structure of *optrA*-positive *E. faecalis* observed in the community, a high degree of homology was observed among *optrA*-positive *E. faecalis* isolates from the community and other sources, including clinics, pigs, chickens, pets, and the environment, suggesting that multiple sources, including animals and clinics, contribute to human *optrA* carriage. The results of this study suggest that the mechanisms for generating *Enterococcus* resistance may be complex, with multiple antibiotic metabolism pathways possible on the basis of multiple resistance phenotypes [27,28,29,30,31].

*OptrA*-positive *Enterococcus* bacteria have been repeatedly reported in clinical, food and livestock samples from Asia, Europe and Latin America. However, the detection rates and plasmid profiles vary across different regions. Comparative data from other regions show variable prevalence of *optrA*-positive *Enterococcus* isolates, ranging from 2 to 8% in Europe to over 15% in some Asian and South American livestock populations. The high detection rate observed in Ningxia is consistent with regions where florfenicol and macrolide use in animal production remains common, suggesting that antibiotic usage patterns and farm management practices are key factors influencing ARG dissemination worldwide [32,33].

The phylogenetic tree of 89 strains of *Enterococcus* can be divided into three branches. The composition of different branch strains is mainly determined by the species, and the strains within the same species have a closer genetic relationship, which is similar to the results of previous studies [34]. Some strains have SNP differences of less than 10, indicating that there have been cross-species transmission events in the Ningxia region. Plasmid-mediated horizontal gene transfer can significantly increase the incidence of ARGs [35]. The risk of possible infection by multidrug-resistant *Enterococcus* from poultry is primarily associated with *E. faecalis*. The extent of this risk remains unknown and appears to be restricted to specific lineages, such as ST16. In branch C1, ST16 is the dominant ST type, which has been reported as an important pathogenic type of *Enterococcus* and can carry a large number of virulence genes. At the same time, it is more likely to form strong or moderate biofilms than other ST types. In previous reports, ST16 is the most common type of *Enterococcus* causing hospital-acquired and linezolid-resistant fecal *Enterococcus* [28].

A total of 52 virulence genes were detected in Ningxia *Enterococcus*. Among them, we found that the hyphal manipulator genes *ebpA*, *ebpB*, and *ebpC*; the genes *fsrA*, *fsrB*, and *fsrC* of the virulence regulator fsr system; and the genes *srtC*, a polymorphism associated with colonization/invasion virulence, and *cpsA*, a biofilm-formation-associated virulence gene, were far more prlent than the other virulence genes [36,37]. These data also emphasize the importance of *Enterococcus* as a repository of virulence genes and the potential for *Enterococcus* to harm humans through the food chain [38]. Combined with the resistance gene results, we found that swine *Enterococcus* carried more virulence and resistance genes compared to other *Enterococcus*. This also suggests that we should be more rational in the use of drugs in swine farming.

However, there are still some limitations in this study, such as the small sample size of sequencing, which may lead to errors in the results. However, the results still have important reference value. Meanwhile, the isolates carried a large number of antibiotic resistance genes and virulence genes, which are seriously harmful to the farming industry. It also suggests the urgency of epidemiologic research and surveillance of *Enterococcus* to prevent future public health risks.

Our findings highlight that swine-derived *Enterococcus* acts as an important reservoir of antimicrobial resistance genes in livestock ecosystems. To mitigate dissemination, we recommend: (1) promoting prudent antibiotic use policies in veterinary medicine; (2) implementing molecular surveillance systems across the farm–environment–clinical interface; (3) improving manure management and biosecurity practices; and (4) integrating antimicrobial-use data into national risk assessment models. Future research should focus on plasmid-mediated transmission dynamics, environmental persistence, and One Health–based intervention strategies.

## 4. Materials and Methods

### 4.1. Source of Samples

A total of 540 strains of *E*. *faecalis*, 214 strains of *E*. *faecium*, and other *Enterococcus* 754 strains were isolated and identified from 25 large-scale farms in various municipalities of Ningxia, China, from 2019 to 2023, using the Merieux Company of France Lyon VITEK 2 COMPACT Fully Automated Microbial Identification System. In total, 89 strains of *Enterococcus* were randomly selected, of which 40 strains were of chicken origin, 28 strains were of bovine origin, 8 strains were of sheep origin, and 13 strains were of porcine origin. Isolation and identification were performed following ISO 6888-1:2018 and ISO 21528-2:2017 guidelines for *Enterococcus* detection using bile esculin agar and VITEK 2 Compact identification [39].

### 4.2. Antimicrobial Susceptibility Testing

*Enterococcus* antimicrobial susceptibility testing, using the Clinical and Laboratory Standards Institute (CLSI) recommended broth microdilution method as the reference to determine the minimum inhibitory concentration [40], was performed on 89 isolates, including 18 antibiotics: Penicillin (P), Amoxicillin/clavulanic acid (A/C), Clindamycin (CLI), Cefoxitin, Cefotiofur (CEF), Erythromycin (EM), Sulfamethoxazole, Trimethoprim/sulfamethoxazole (SXT), Telmicin (TIL), Enrofloxacin (ENR), Ofloxacin (OFL), Vancomycin (VAN), doxycycline (DOX), Gentamicin, Tymiocin (TIA), Florfenicol (FFC), Linezolid (LZD), oxacillin (OXA). *Staphylococcus aureus* ATCC 29213 and *E. faecalis* ATCC 29212 were used as quality control strains. For antibiotics not included in the CLSI guidelines, such as enrofloxacin and florfenicol, interpretation criteria were based on published veterinary studies to reflect their practical importance in animal husbandry [41]. Data were analyzed using one-way ANOVA followed by Tukey’s HSD test for mean separation at the 0.05 significance level. Different letters above the bars indicate statistically significant differences among groups.

### 4.3. Whole Genome Sequencing and Results Analysis

To further investigate the genetic characteristics of the 89 *Enterococcus* strains in this trial, the bacterial genome was extracted using a Genomic DNA extraction kit (OMEGA, Tianjin, China). Short sequences were obtained using NovaSeq 6000 (2 × 150 bp-paired end reads) system from Illumina, San Diego, CA, USA. The quality control of the raw sequencing data was initially checked using FastQC v0.11.9, and then fastp v0.23.2 was used to remove adapters and perform quality trimming (retaining bases with Q ≥ 20 and reads with a length ≥ 50 bp). The average sequencing depth of each sample was ≥30×, and ≥80% of the bases reached Q30. SPAdes 3.15.5 was used to assemble the sequencing data [42]. Subsequent data analysis was performed using ResFinder 4.1 (https://cge.food.dtu.dk/services/ResFinder/, accessed on 18 April 2025) for prediction of acquired AMR genes (90% concordance and 60% coverage), VirulenceFinder (https://cge.food.dtu.dk/services/VirulenceFinder/ accessed on 18 April 2025) for prediction of virulence genes (90% concordance and 60% coverage), PlasmidFinder (https://cge.food.dtu.dk/services/PlasmidFinder/ accessed on 18 April 2025) for identification of plasmid replicons (95% concordance and 60% coverage). MLST types were analyzed using (https://cge.food.dtu.dk/services/MLST/ accessed on 18 April 2025) for MLST types. All of the aforementioned websites were accessed on 1 May 2024. Correlation of resistance genes with plasmids was analyzed using Cytoscape_v3.7.0 software [43,44,45].

### 4.4. SNP Phylogenetic Analysis

Phylogenetic tree analysis of the 89 *Enterococcus* producing strains sequenced in this study. The genome was first annotated using Prokka 1.2.0, the core genome alignment was obtained using the Roary program3.11.2, and the maximum likelihood tree was assembled using the FastTree program2.1.10. Visualization and annotation were performed through iTOL (https://itol.embl.de, accessed on 18 April 2025) for the beautification of the evolutionary tree. Finally, the SVG format picture of the evolution tree was imported into the AI software to adjust the format [46,47].

### 4.5. Statistical Analysis

All statistical analyses were performed using GraphPad Prism 9.0 (GraphPad Software, San Diego, CA, USA). Data were expressed as mean ± standard deviation (SD). Differences in antimicrobial resistance rates among animal species were evaluated using one-way ANOVA, followed by Tukey’s HSD test for mean separation. Statistical significance was determined at *p* < 0.05.

## 5. Conclusions

In summary, this study used WGS combined with biological analysis methods to provide accurate information on the antibiotic resistance status and pathogenicity of *Enterococcus* of multiple animal origins in Ningxia region of China. *E. faecalis* was the predominant species among animal-derived *Enterococcus* in Ningxia, exhibiting high resistance to clindamycin, cefoxitin, and sulfamethoxazole. The frequent detection of *optrA* on rep9-family plasmids in pig isolates highlights swine as an important reservoir of linezolid resistance genes with potential zoonotic risk.

## Figures and Tables

**Figure 1 antibiotics-14-01137-f001:**
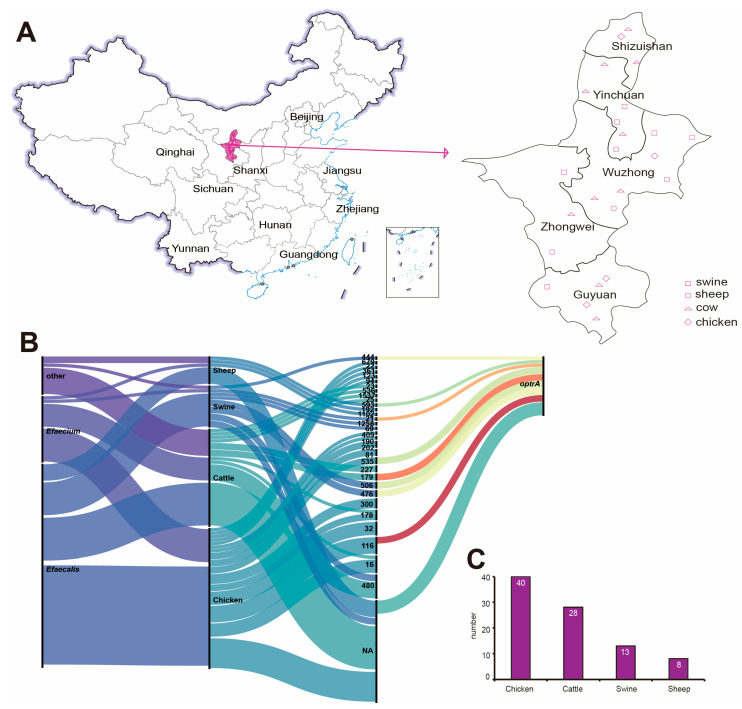
*Enterococcus* sampling map and basic information statistics. (**A**) *Enterococcus* sampling map. (**B**) Sankey map of genetic information of *Enterococcus*. (**C**) *Enterococcus* host statistics histogram.

**Figure 2 antibiotics-14-01137-f002:**
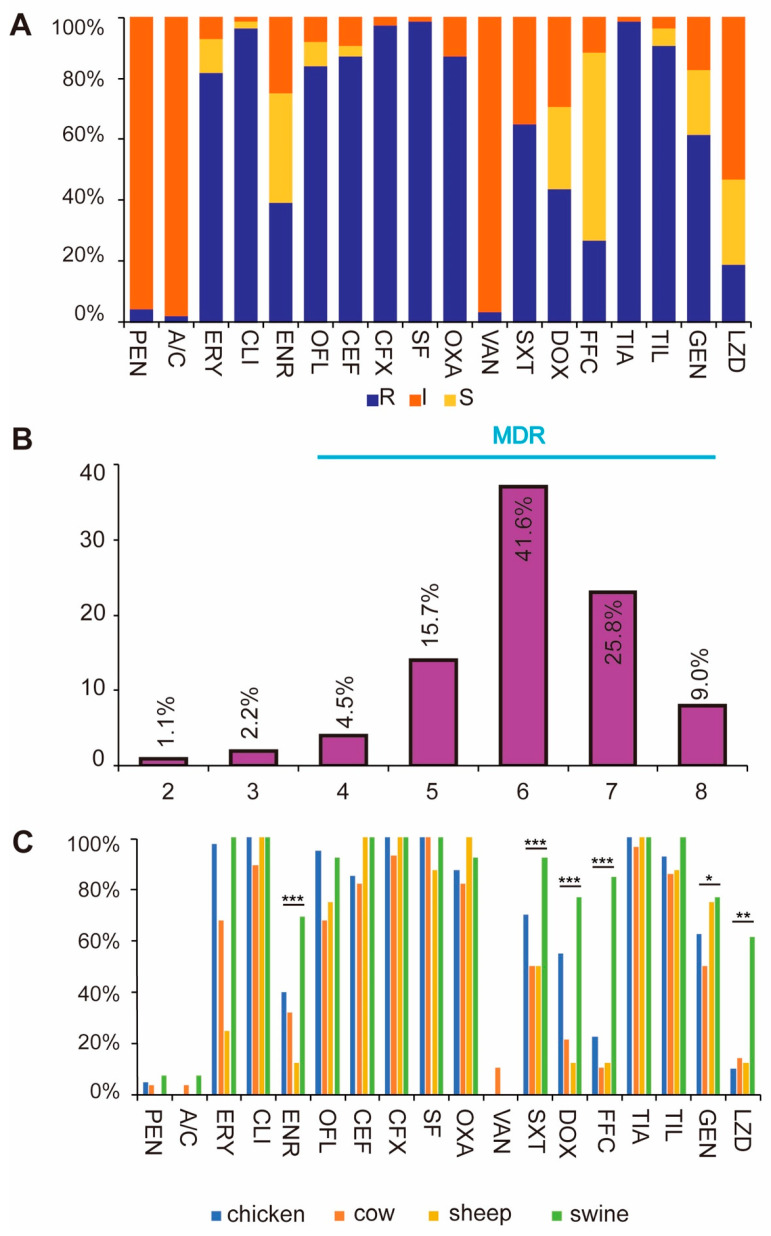
Antibiotic susceptibility results. Figure (**A**) presents the overall antibiotic resistance results of *Enterococcus*. S indicates sensitive, I indicates moderately sensitive, and R indicates resistant. Figure (**B**) shows multi drug-resistant strains. Figure (**C**) presents the antibiotic resistance situations of different types of *Enterococcus* (* *p* ≤ 0.05, ** *p* ≤ 0.01, *** *p* ≤ 0.001. *p* < 0.05, ANOVA followed by Tukey’s HSD test).

**Figure 3 antibiotics-14-01137-f003:**
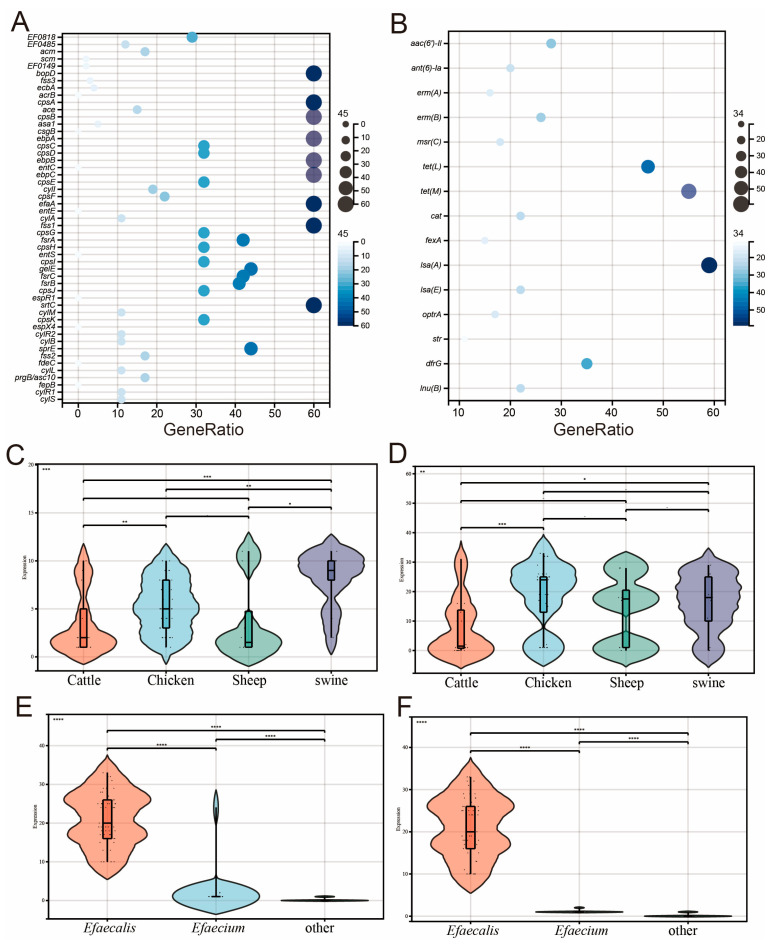
Genetic characteristic information. (**A**) Bubble plot showing the carriage status of virulence genes. (**B**) Bubble plot depicting the carriage status of drug resistance genes. (**C**) Diagram of the differences in the carriage of drug resistance genes among different species. (**D**) Diagram of the differences in the carriage of virulence genes among different species. (**E**) Diagram of the differences in the carriage of drug resistance genes among different taxa. (**F**) Diagram of the differences in the carriage of virulence genes among different taxa. (* *p* ≤ 0.05, ** *p* ≤ 0.01, *** *p* ≤ 0.001, **** *p* ≤ 0.0001).

**Figure 4 antibiotics-14-01137-f004:**
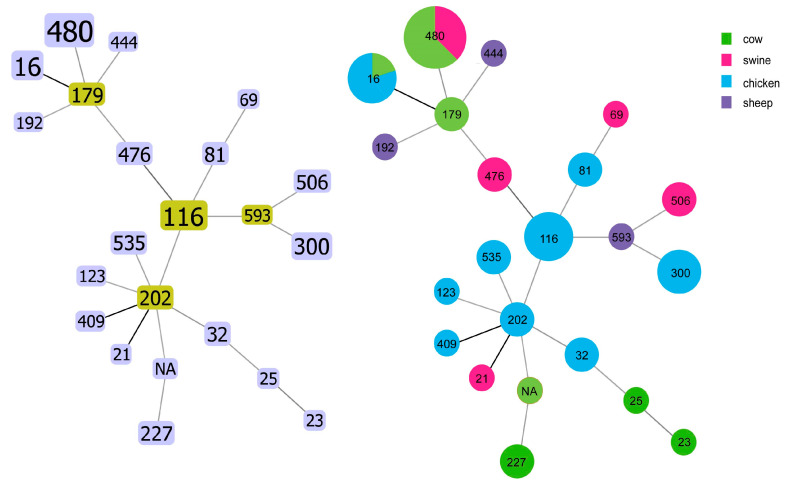
*E. faecalis* MLST phylogenetic tree.

**Figure 5 antibiotics-14-01137-f005:**
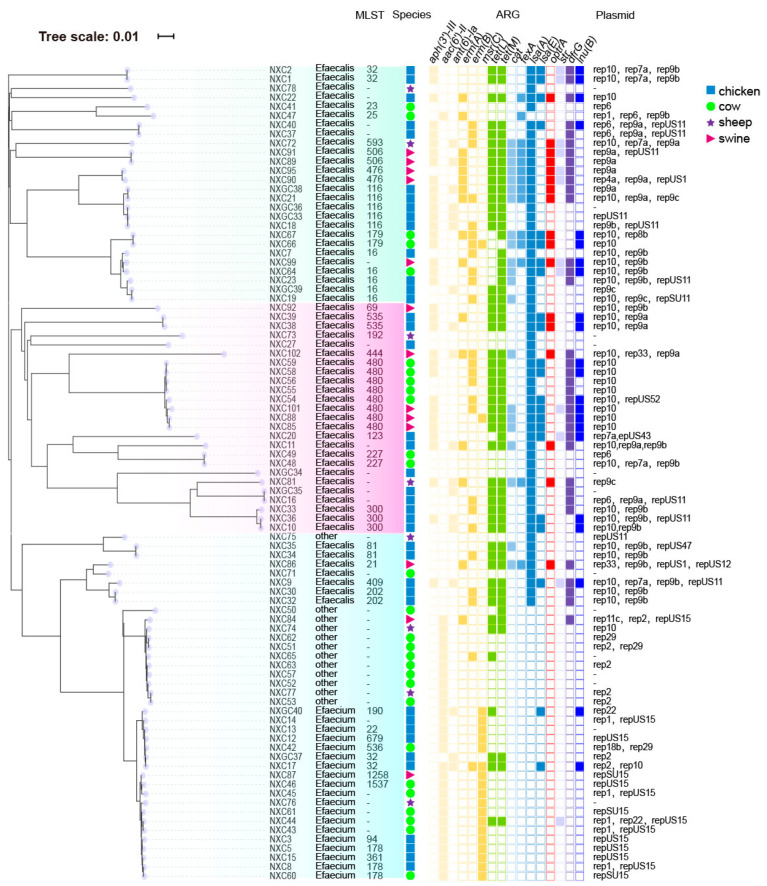
Phylogenetic tree of *Enterococcus* SNPS. From left to right, phylogenetic tree, MLST typing, species, resistance gene, and carrying plasmid. The whole thing is divided into three branches: C1 in green, C2 in pink and C3 in blue.

**Figure 6 antibiotics-14-01137-f006:**
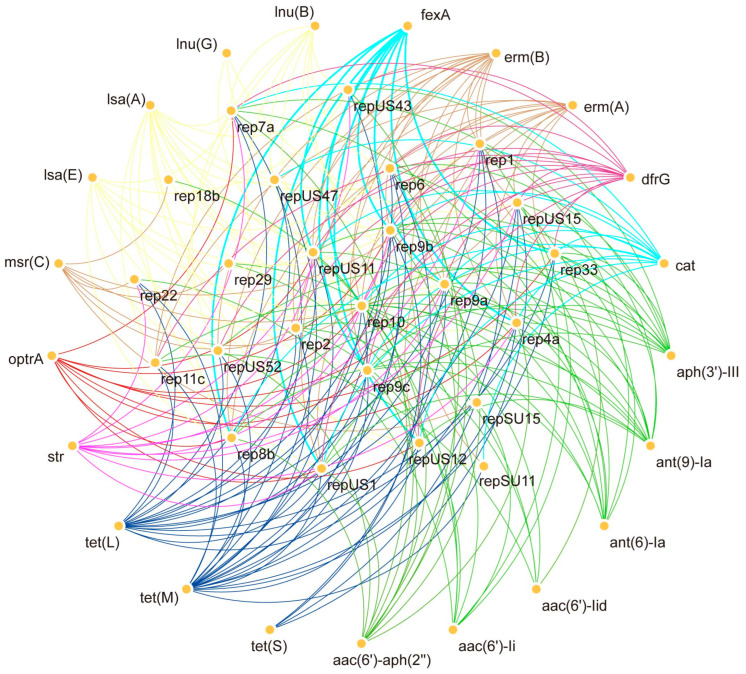
Bar graph of Coexistence network of plasmids and antibiotics resistance genes carried by *Enterococcus*.

**Table 1 antibiotics-14-01137-t001:** Overall Antibiotics susceptibility results of *Enterococcus*.

Antibiotic Type Drug	Name	Overall Sensitivity		
		R	I	S
Penicillin	Benzylpenicillin	87.6%	0	12.4%
	Penicillin	4.5%	0	95.5%
	Augmentin	2.2%	0	97.8%
Clindamycin	Clindamycin	96.6%	2.2%	1.1%
Quinolones	Enrofloxacin	39.3%	36.0%	24.7%
	Ofloxacin	84.3%	7.9%	7.9%
beta-lactams	Ceftiofur	87.6%	3.4%	9.0%
	Cefoxitin	97.8%	0	2.2%
Sulfonamides	Sulfamethoxazole	98.9%	0	1.1%
Glycopeptides	Vancomycin	3.4%	0	96.6%
Sulfonamides	Cotrimoxazole	65.2%	0	36.0%
Tetracycline	Doxycycline	43.8%	27.0%	29.2%
	Florfenicol	27.0%	61.8%	11.2%
Macrolides	Erythromycin	82.0%	11.2%	6.7%
	Taimiocin	98.9%	0	1.1%
	Temicosin	91.0%	5.6%	3.4%
Aminoglycosides	Gentamycin	61.8%	21.3%	16.9%
Oxazolidinones	Linezolid	19.1%	28.1%	52.8%

Note: S, sensitive; I, moderately sensitive; R, drug resistance.

## Data Availability

The complete genome sequences of 89 *Enterococcus* isolates have been deposited in the GenBank database under accession number: PRJNA1130870.

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
