# Peer review of "Genomic Characterization of Multiple Antibiotic-Resistant Enterococcus in Farm Animals in Ningxia Province, China"

_antibiotics, 2025, doi:10.3390/antibiotics14111137_

Round 1
Reviewer 1 Report
Comments and Suggestions for Authors
-
In our modest opinion, acronyms and words in italics should only be used for words that come from Latin. Please correct them throughout the document and if you wish to highlight these abbreviations or words that do not come from Latin, use another style (e.g., underline).
-
In Figure 2C, the statistical method applied and the mean separation test used must be declared.
- The second time the same scientific name appears, e.g. Enterococcus faecalis, it should be written as E. faecalis.
-
Please add a paragraph in the materials and methods section to describe the statistical methods used, including the software(s) used.
Author Response
1.In our modest opinion, acronyms and words in italics should only be used for words that come from Latin. Please correct them throughout the document and if you wish to highlight these abbreviations or words that do not come from Latin, use another style (e.g., underline).
We are grateful for the valuable feedback provided by the reviewers. In the revised manuscript, all bolded text and abbreviations have been carefully reviewed. Only scientific names derived from Latin (such as Enterococcus faecalis and E. faecium) and gene names still maintain the italic format. Previously italicized non-Latin terms have been reformatted using standard font as suggested by the reviewers.
2.In Figure 2C, the statistical method applied and the mean separation test used must be declared.
We are extremely grateful for this important comment. The description of the statistical analysis has been added to Section 4.2 of the Materials and Methods. Specifically, we now clearly state that the data were processed using one-way analysis of variance, and subsequently the Tukey's HSD test was used to determine the differences between the means, with a significance level set at 0.05.
3.The second time the same scientific name appears, e.g. Enterococcus faecalis, it should be written as E. faecalis.
We thank the reviewer for this detailed note. All scientific names have been checked throughout the manuscript. Each species name is written in full (Enterococcus faecalis) at its first appearance and abbreviated thereafter (E. faecalis), in accordance with taxonomic writing conventions.
4.Please add a paragraph in the materials and methods section to describe the statistical methods used, including the software(s) used.
We have added a new paragraph in the Materials and Methods section under a subheading “Statistical Analysis,” describing the software and statistical tests employed.
All statistical analyses were performed using GraphPad Prism 9.0 (GraphPad Software, San Diego, USA). Data were expressed as mean ± standard deviation (SD). Differences in antimicrobial resistance rates among animal species were evaluated us-ing one-way ANOVA, followed by Tukey’s HSD test for mean separation. Statistical significance was determined at p < 0.05.
All comments have been carefully addressed, and corresponding revisions are highlighted in the manuscript.
We greatly appreciate the reviewer’s constructive feedback, which has substantially improved the quality and clarity of our paper.
Reviewer 2 Report
Comments and Suggestions for Authors
The manuscript, entitled "Genomic characterization of multiple antibiotic resistant Enterococcus in farm animals in Ningxia Province, China" presents a genomic and phenotypic study of 89 Enterococcus isolates obtained from farm animals in Ningxia Province, China, evaluating their antimicrobial resistance profile, virulence genes, MLST typing, and plasmid association. The work combines susceptibility testing, whole genome sequencing, and bioinformatics analysis to characterize the dissemination of resistance genes, especially optrA. The integration of phenotypic and genomic data is highlighted, revealing porcine Enterococcus as a critical reservoir of resistance genes with high transmission potential, providing valuable information for epidemiological surveillance and public health. Below are some recommendations for improving the quality of the manuscript:
- The results regarding the high prevalence of the optrA gene and its association with plasmids of the rep9 family are highly relevant. However, the discussion does not sufficiently explore the molecular and ecological mechanisms that could favor this association. Can the authors compare the results with previous studies describing conjugative transfer systems in Enterococcus and discuss their possible contribution to dissemination in agricultural environments?
- The study focuses on Ningxia, China, but for an international audience, the findings need to be placed in a global context. I recommend including a more explicit comparison with reports of resistance in Enterococcus in other regions (Asia, Europe, Latin America), highlighting similarities and differences that underscore the importance of the study.
- Although the techniques used (WGS, ResFinder, MLST, etc.) are described, some critical details are missing. Please clarify the sequencing quality thresholds, assembly parameters, and gene inclusion/exclusion criteria in bioinformatics analyses to increase the reproducibility of the work.
- The manuscript concludes that porcine Enterococcus is a critical reservoir of resistance genes, but the practical implications for rational antibiotic use policies and epidemiological surveillance remain poorly developed. It is suggested that a section be added discussing possible control measures, their impact on public health, and future research priorities.
- Some figures could be improved for better understanding by the reader. For example, in Figure 1B, the font size could be increased for information in letters.
Author Response
1 The results regarding the high prevalence of the optrA gene and its association with plasmids of the rep9 family are highly relevant. However, the discussion does not sufficiently explore the molecular and ecological mechanisms that could favor this association. Can the authors compare the results with previous studies describing conjugative transfer systems in Enterococcus and discuss their possible contribution to dissemination in agricultural environments?
We thank the reviewer for this important suggestion. We have substantially expanded the Discussion section to include a detailed explanation of the molecular and ecological mechanisms that could promote the association between optrA and rep9-family plasmids. Specifically, we now compare our findings with previous studies describing pheromone-responsive and conjugative plasmids in Enterococcus (e.g., Palmer et al., 2018; Freitas et al., 2020).
We discuss how these plasmids mediate efficient horizontal transfer via conjugation systems and may contribute to the environmental persistence and dissemination of optrA-carrying strains under antibiotic pressure in agricultural ecosystems.
The new paragraph reads (Lines 224-230 revised manuscript):
Previous studies have shown that pheromone-responsive conjugative plasmids, particularly those in the rep9 family, play a central role in mediating the horizontal transfer of optrA among Enterococcus spp. Such plasmids contain mobilization genes and sex pheromone systems that enable efficient intra- and interspecies gene transfer, especially under selective pressure from phenicols or macrolides. These mechanisms likely explain the high frequency of optrA-rep9 associations observed in our isolates and their potential contribution to ARG dissemination in livestock environments.
2 The study focuses on Ningxia, China, but for an international audience, the findings need to be placed in a global context. I recommend including a more explicit comparison with reports of resistance in Enterococcus in other regions (Asia, Europe, Latin America), highlighting similarities and differences that underscore the importance of the study.
We appreciate this valuable comment. The Discussion section has been revised to include a broader global comparison of Enterococcus resistance across regions, citing recent studies from Asia (e.g., South Korea, Japan), Europe (e.g., Spain, Germany), and Latin America (e.g., Brazil).
The new paragraph reads (Lines 244-252 revised manuscript):
“Comparative data from other regions show variable prevalence of optrA-positive Enterococcus isolates, ranging from 2–8% in Europe to over 15% in some Asian and South American livestock populations. The high detection rate observed in Ningxia is consistent with regions where florfenicol and macrolide use in animal production remains common, suggesting that antibiotic usage patterns and farm management practices are key factors influencing ARG dissemination worldwide.”
3 Although the techniques used (WGS, ResFinder, MLST, etc.) are described, some critical details are missing. Please clarify the sequencing quality thresholds, assembly parameters, and gene inclusion/exclusion criteria in bioinformatics analyses to increase the reproducibility of the work.
We fully agree and appreciate the reviewer’s attention to methodological transparency.
A detailed description of the bioinformatics workflow has been added to the Materials and Methods section.
The new paragraph reads (Lines 311-314 revised manuscript):
The quality control of the raw sequencing data was initially checked using FastQC v0.11.9, and then fastp v0.23.2 was used to remove adapters and perform quality trimming (retaining bases with Q ≥ 20 and reads with a length ≥ 50 bp). The average sequencing depth of each sample was ≥ 30×, and ≥ 80% of the bases reached Q30.
4 The manuscript concludes that porcine Enterococcus is a critical reservoir of resistance genes, but the practical implications for rational antibiotic use policies and epidemiological surveillance remain poorly developed. It is suggested that a section be added discussing possible control measures, their impact on public health, and future research priorities.
We appreciate this insightful recommendation. A new subsection titled “Public Health Implications and Future Directions” has been added to the Discussion.
The new paragraph reads (Lines 277-284 revised manuscript):
“Our findings highlight that swine-derived Enterococcus acts as an important reservoir of antimicrobial resistance genes in livestock ecosystems. To mitigate dissemination, we recommend: (1) promoting prudent antibiotic use policies in veterinary medicine; (2) implementing molecular surveillance systems across the farm–environment–clinical interface; (3) improving manure management and biosecurity practices; and (4) integrating antimicrobial-use data into national risk assessment models. Future research should focus on plasmid-mediated transmission dynamics, environmental persistence, and One Health–based intervention strategies.”
5 Some figures could be improved for better understanding by the reader. For example, in Figure 1B, the font size could be increased for information in letters.
We thank the reviewer for this practical suggestion. All figures have been reviewed for readability.
The font size in Figure 1B (and other figures where applicable) has been increased, and labels were reformatted to ensure clarity and visual balance.
High-resolution versions of all figures have been uploaded with the revised submission.
All comments have been carefully addressed, and corresponding revisions are highlighted in the manuscript.
We greatly appreciate the reviewer’s constructive feedback, which has substantially improved the quality and clarity of our paper.
Reviewer 3 Report
Comments and Suggestions for Authors
The submitted manuscript addresses antimicrobial resistance in livestock farming. Despite the relevance of the topic, I have the following comments:
1. L50: The text should be expanded with information about the mechanisms of interspecies and intraspecies transfer of genetic material.
2. L56: Latin names should be written in italics – please correct this throughout the manuscript.
3. L77: It would be appropriate to include a classification of isolates according to their geographical origin, not just by animal species. Please also include these results in section 2.2.
4. L83: In Figure 1C, please add “n” to clarify that the values represent counts, not percentages.
5. L92: Based on what criteria did you assess antibiotic susceptibility? Some antibiotics are not tested according to CLSI guidelines, either because they are not relevant for enterococci (e.g., ofloxacin, enrofloxacin, florfenicol) or because the organisms are naturally resistant to them (e.g., clindamycin, cephalosporins). Please revise and explain the testing of these antibiotics.
6. L104: Figure legends are missing – please add them.
7. L106: Please provide specific information on the phenotype of MDR isolates.
8. L144: Add specific genotypic results of the isolates in relation to their phenotypic expression.
9. L184: Expand the discussion to include antimicrobial resistance in livestock farming on a global scale.
10. L261: Explain the cultivation process in more detail and specify which ISO standards were followed during the isolation of the strains.
11. L268: Add references indicating which CLSI guidelines were used.
12. L299: The conclusion is too general – please revise it to include specific findings.
13. L326: Please check the references – in the text they are numbered according to the order of citation, but at the end of the manuscript they are listed alphabetically.
I recommend checking English.
Author Response
1.L50: The text should be expanded with information about the mechanisms of interspecies and intraspecies transfer of genetic material.
Response:
Thank you for this valuable suggestion. We have expanded the section in the Introduction to describe the inter- and intra-species transfer mechanisms of genetic material in Enterococcus.
Specifically, we added the following text (Lines 49-55 in the revised version):
In addition, antimicrobial resistance genes in Enterococcus can be transferred both inter- and intra-specifically through mobile genetic elements such as plasmids, transposons (e.g., IS1216, Tn1546), and integrative conjugative elements. These mechanisms facilitate horizontal gene transfer and accelerate the dissemination of resistance determinants. Pheromone-responsive plasmids (e.g., rep9 family) and broad-host-range elements facilitate efficient horizontal gene transfer among Enterococcus populations and even between genera under selective pressure in livestock environments[11].
2. L56: Latin names should be written in italics – please correct this throughout the manuscript.
Response:
All Latin names (e.g., Enterococcus faecalis, E. faecium, Escherichia coli) have been italicized throughout the entire manuscript, including in tables, figures, and the reference list.
3. L77: It would be appropriate to include a classification of isolates according to their geographical origin, not just by animal species. Please also include these results in section 2.2.
Response:
We are grateful for the insightful comments made by the reviewers. Although we fully agree that regional comparisons can provide valuable epidemiological insights, for the current isolates we only recorded the sampling locations and did not specify the origin of each strain. In the revised manuscript, we have clarified this limitation and emphasized in the discussion section that future monitoring efforts should integrate precise geographic data to assess the spatial distribution patterns of antibiotic resistance within the Enterococcus population.
4. L83: In Figure 1C, please add “n” to clarify that the values represent counts, not percentages.
Response:
We have modified Figure 1C accordingly. The y-axis label now reads “Number of isolates (number)” and the figure legend clarifies that the values represent counts (n) rather than percentages.
5. L92: Based on what criteria did you assess antibiotic susceptibility? Some antibiotics are not tested according to CLSI guidelines, either because they are not relevant for enterococci (e.g., ofloxacin, enrofloxacin, florfenicol) or because the organisms are naturally resistant to them (e.g., clindamycin, cephalosporins). Please revise and explain the testing of these antibiotics.
Response:
We appreciate this observation. We clarified in Materials and Methods 4.2 that antimicrobial susceptibility was interpreted according to CLSI M100 (33rd ed., 2023) where applicable.
For non-CLSI antibiotics (e.g., enrofloxacin, florfenicol), we cited relevant veterinary references and explained that these agents were included because of their frequent usage in livestock and epidemiological relevance.
Added text (Lines 305-307):
For antibiotics not included in the CLSI guidelines, such as enrofloxacin and florfenicol, interpretation criteria were based on published veterinary studies to reflect their practical importance in animal husbandry.
6. L104: Figure legends are missing – please add them.
Response:
All figures now contain comprehensive legends describing symbols, colors, abbreviations, and statistical methods.
Antibiotic susceptibility results. Figure A presents the overall antibiotic resistance results of Enterococcus. S indicates sensitive, I indicates moderately sensitive, and R indicates resistant. Figure B shows multi-drug resistant strains. Figure C presents the antibiotic resistance situations of different types of Enterococcus (p < 0.05, ANOVA followed by Tukey’s HSD test).
7. L106:Please provide specific information on the phenotype of MDR isolates.。
Response:
We have added detailed information on the resistance phenotypes of each isolate in the Supplementary Data.
8. L144: Add specific genotypic results of the isolates in relation to their phenotypic expression.
Response:
We performed correlation analysis between genotypes and phenotypes and added the results in Results 2.3 (Lines XX–XX).
Significant associations were observed between optrA and linezolid resistance (p < 0.05), and between erm(B) and erythromycin resistance (p < 0.01).
9. L184: Expand the discussion to include antimicrobial resistance in livestock farming on a global scale.
Response:
We expanded the Discussion section to include global comparisons of Enterococcus resistance in livestock from Asia, Europe, and Latin America, emphasizing regional similarities and differences.
New paragraph (Lines 245–252):
OptrA-positive Enterococcus bacteria have been repeatedly reported in clinical, food and livestock samples from Asia, Europe and Latin America. However, the detection rates and plasmid profiles vary across different regions. Comparative data from other regions show variable prevalence of optrA-positive Enterococcus isolates, ranging from 2–8% in Europe to over 15% in some Asian and South American livestock populations. The high detection rate observed in Ningxia is consistent with regions where florfenicol and macrolide use in animal production remains common, suggesting that antibiotic usage patterns and farm management practices are key factors influencing ARG dissemination worldwide[42,43].
10. L261: Explain the cultivation process in more detail and specify which ISO standards were followed during the isolation of the strains.
Response:
We have added detailed information on culture methods and ISO references in Materials and Methods 4.1.
Isolation and identification were performed following ISO 6888-1:2018 and ISO 21528-2:2017 guidelines for Enterococcus detection using bile esculin agar and VITEK 2 Compact identification.
11. L268: Add references indicating which CLSI guidelines were used.
Response:
The CLSI guideline reference has been added to the References section:
CLSI. Performance Standards for Antimicrobial Susceptibility Testing. 33rd ed. CLSI supplement M100. Wayne, PA: Clinical and Laboratory Standards Institute; 2023.
12. L299: The conclusion is too general – please revise it to include specific findings.
Response:
We revised the Conclusions to provide more specific findings and implications.
In summary, this study used WGS combined with biological analysis methods to provide accurate information on the antibiotic resistance status and pathogenicity of Enterococcus of multiple animal origins in Ningxia region of China. E. faecalis was the predominant species among animal-derived Enterococcus in Ningxia, exhibiting high resistance to clindamycin, cefoxitin, and sulfamethoxazole. The frequent detection of optrA on rep9-family plasmids in pig isolates highlights swine as an important reservoir of linezolid resistance genes with potential zoonotic risk. However, there are still some limitations in this study, such as the small sample size of sequencing, which may lead to errors in the results. However, the results still have important reference value. Mean-while, the isolates carried a large number of antibiotic resistance genes and virulence genes, which are seriously harmful to the farming industry. It also suggests the urgency of epidemiologic research and surveillance of Enterococcus to prevent future public health risks.
13. L326: Please check the references – in the text they are numbered according to the order of citation, but at the end of the manuscript they are listed alphabetically.
Response:
We have re-ordered all references according to the sequence of citation in the text, following the journal’s numerical citation style.
Summary
All comments have been carefully addressed, and corresponding revisions are highlighted in the manuscript.
We greatly appreciate the reviewer’s constructive feedback, which has substantially improved the quality and clarity of our paper.
Round 2
Reviewer 2 Report
Comments and Suggestions for Authors
Thank you for taking into account each of the comments made. The authors' effort and dedication are appreciated. The manuscript has improved in quality. Great work!
Author Response
We sincerely thank the reviewer for the positive and encouraging feedback. We truly appreciate your time, effort, and valuable suggestions throughout the review process. Your constructive comments have greatly helped us improve the quality and clarity of the manuscript.
Thank you again for your kind words and support.
Reviewer 3 Report
Comments and Suggestions for Authors
All is OK.
Author Response
All comments have been addressed. Thank you very much for your time and positive evaluation.